# Optimal Neural Network Approximation of Wasserstein Gradient Direction via Convex Optimization

## Abstract

The computation of Wasserstein gradient direction is essential for posterior sampling problems and scientific computing. The approximation of the Wasserstein gradient with finite samples requires solving a variational problem. We study the variational problem in the family of two-layer networks with squared-ReLU activations, towards which we derive a semi-definite programming (SDP) relaxation. This SDP can be viewed as an approximation of the Wasserstein gradient in a broader function family including two-layer networks. By solving the convex SDP, we obtain the optimal approximation of the Wasserstein gradient direction in this class of functions. We also propose practical algorithms using subsampling and dimension reduction. Numerical experiments including PDE-constrained Bayesian inference and parameter estimation in COVID-19 modeling demonstrate the effectiveness and efficiency of the proposed method.

## 1 Introduction

Bayesian inference plays an essential role in learning model parameters from the observational data with applications in inverse problems, scientific computing, information science, and machine learning (Stuart, 2010). The central problem in Bayesian inference is to draw samples from a posterior distribution, which characterizes the parameter distribution given data and a prior distribution.

The Wasserstein gradient flow (Otto, 2001; Ambrosio et al., 2005; Junge et al., 2017) has shown to be effective in drawing samples from a posterior distribution, which attracts increasing attention in recent years. For instance, the Wasserstein gradient flow of Kullback-Leibler (KL) divergence connects to the overdampled Langevin dynamics. The time-discretization of the overdamped Langevin dynamics renders the classical Langevin Monte Carlo Markov Chain (MCMC) algorithm. In this sense, the computation of Wasserstein gradient flow yields a different viewpoint for sampling algorithms. In particular, the Wasserstein gradient direction also provides a deterministic update of the particle system (Carrillo et al., 2021b). Based on the approximation or generalization of the Wasserstein gradient direction, many efficient sampling algorithms have been developed, including Wasserstein gradient descent (WGD) with kernel density estimation (KDE) (Liu et al., 2019), Stein variational gradient descent (SVGD) (Liu & Wang, 2016), and neural variational gradient descent (di Langosco et al., 2021), etc.

Meanwhile, neural networks exhibit tremendous optimization and generalization performance in learning complicated functions from data. They also have wide applications in Bayesian inverse problems (Rezende & Mohamed, 2015; Onken et al., 2020; Kruse et al., 2019; Lan et al., 2021). According to the universal approximation theorem of neural networks (Hornik et al., 1989; Lu et al., 2017), any arbitrarily complicated functions can be learned by a two-layer neural network with nonlinear activations and a sufficient number of neurons. Functions represented by neural networks naturally provide an approximation towards the Wasserstein gradient direction.

However, due to the nonlinear and nonconvex structure of neural networks, optimization algorithms including stochastic gradient descent may not find the global optima of the training problem. Recently, based on a line of works (Pilanci & Ergen, 2020; Sahiner et al., 2020; Bartan & Pilanci, 2021a), the regularized training problem of two-layer neural networks with ReLU/polynomial activation can be formulated as a convex program. Indeed, by solving the convex program, we can

construct the entire set of global optima of the nonconvex training problem (Wang et al., 2020). Theoretical analysis (Wang et al., 2022) shows that global optima of the training problem correspond to the simplest models with good generalization properties. Moreover, numerical results (Pilanci & Ergen, 2020) show that neural networks found by solving the convex program can achieve higher train accuracy and test accuracy compared to neural networks trained by SGD with the same number of parameters.

In this paper, we study a variational problem, whose optimal solution corresponds to the Wasserstein gradient direction. Focusing on the family of two-layer neural networks with squared ReLU activation, we formulate the regularized variational problem in terms of samples. Directly training the neural network to minimize the loss may get the neural network stuck at local minima or saddle points and it often leads to biased sample distribution from the posterior. Instead, we analyze the convex dual problem of the training problem and study its semi-definite program (SDP) relaxation by analyzing the geometry of dual constraints. The resulting SDP can be efficiently solved by convex optimization solvers such as CVXPY (Diamond & Boyd, 2016). We then derive the corresponding relaxed bidual problem (dual of the relaxed dual problem). Thus, the optimal solution to the dual problem yields an optimal approximation of the Wasserstein gradient direction in a broader function family. We also analyze the choice of the regularization parameter and present a practical implementation using subsampling and parameter dimension reduction to improve computational efficiency. Numerical results for experiments including PDE-constrained inference problems and Covid-19 parameter estimation problems illustrate the effectiveness and efficiency of our method.

## 1.1 RELATED WORKS

The time and spatial discretizations of Wasserstein gradient flows are extensively studied in literature (Jordan et al., 1998; Junge et al., 2017; Carrillo et al., 2021a;b; Bonet et al., 2021; Liutkus et al., 2019; Frogner & Poggio, 2020). Recently, neural networks have been applied in solving or approximating Wasserstein gradient flows (Mokrov et al., 2021; Lin et al., 2021b;a; Alvarez-Melis et al., 2021; Bunne et al., 2021; Hwang et al., 2021; Fan et al., 2021). For sampling algorithms, di Langosco et al. (2021) learns the transportation function by solving an unregularized variational problem in the family of vector-output deep neural networks. Compared to these studies, we focus on a convex SDP relaxation of the varitional problem induced by the Wasserstein gradient direction. Meanwhile, Feng et al. (2021) form the Wasserstein gradient direction as the mininimizer the Bregman score and they apply deep neural networks to solve the induced variational problem.

In comparison to previous works on the convex optimization formulations of neural networks using SDP (Bartan & Pilanci, 2021a;b), they focus on the polynomial activation and give the exact convex optimization formulation (instead of convex relaxation). In comparison, we focus on the neural networks with the squared ReLU activation, which has not been considered before. Our method can also apply to the analysis of supervised learning problem using squared ReLU activated neural networks.

## 2 BACKGROUND

In this section, we briefly review the Wasserstein gradient descent and present its variational formulation. In particular, we focus on the Wasserstein gradient descent direction of KL divergence functional. Later on, we design a neural network convex optimization problem to approximate the Wasserstein gradient in samples.

### 2.1 WASSERSTEIN GRADIENT DESCENT

Consider an optimization problem in the probability space:

$$\inf_{\rho \in \mathcal{P}} D_{KL}(\rho \| \pi) = \int \rho(x)(\log \rho(x) - \log \pi(x))dx, \tag{1}$$

Here the integral is taken over $\mathbb{R}^d$ and the objective functional $D_{KL}(\rho \| \pi)$ is the KL divergence from $\rho$ to $\pi$. The variable is the density function $\rho$ in the space $\mathcal{P} = \{\rho \in C^\infty(\mathbb{R}^d) | \int \rho dx = 1, \ \rho > 0\}$. The function $\pi \in C^\infty(\mathbb{R}^d)$ is a known probability density function of the posterior distribution. By solving the optimization problem (1), we can generate samples from the posterior distribution.

A known fact (Villani, 2003, Chapter 8.3.1) is that the Wasserstein gradient descent flow for the optimization problem (1) satisfies

$$\partial_t \rho_t = \nabla \cdot \left( \rho_t \nabla \frac{\delta}{\delta \rho_t} D_{KL}(\rho_t \| \pi) \right) = \nabla \cdot (\rho_t (\nabla \log \rho_t - \nabla \log \pi))$$

$$\overset{(a)}{=} \Delta \rho_t - \nabla \cdot (\rho_t \nabla \log \pi),$$

where $\rho_t(x) = \rho(x,t)$, $\frac{\delta}{\delta \rho_t}$ is the $L^2$ first variation operator w.r.t. $\rho_t$, $\nabla \cdot F$ denotes the divergence of a vector valued function $F : \mathbb{R}^d \to \mathbb{R}^d$ and $\Delta$ is the Laplace operator. In step (a) we uses the fact that $\rho_t \nabla \log \rho_t = \nabla \rho_t$. This equation is also known as the gradient drift Fokker-Planck equation. It corresponds to the following updates in terms of samples:

$$dx_t = -(\nabla \log \rho_t(x_t) - \nabla \log \pi(x_t))dt, \tag{2}$$

where $x_t$ follows the distribution of $\rho_t$. Clearly, when $\rho_t = \pi$, the above dynamics reach the equilibrium, which implies that the samples $x_t$ are generated by the posterior distribution.

To solve the Wasserstein gradient flow (2), we consider a forward Eulerian discretization in time. In the $l$-th iteration, suppose that $\{x_l^n\}$ are samples drawn from $\rho_l$. The update rule of Wasserstein gradient descent (WGD) on the particle system $\{x_l^n\}$ follows

$$x_{l+1}^n = x_l^n - \alpha_l \nabla \Phi_l(x_l^n), \tag{3}$$

where $\Phi_l : \mathbb{R}^d \to \mathbb{R}$ is a function which approximates $\log \rho_l - \log \pi$ and $\alpha_l > 0$ is the step size.

## 2.2 Variational formulation of WGD

Given the particles $\{x_n\}_{n=1}^N$, we design the following variational problem to choose a suitable function $\Phi$ approximating the function $\log \rho - \log \pi$. Consider

$$\inf_{\Phi \in C^1(\mathbb{R}^d)} \frac{1}{2} \int \|\nabla \Phi(x - (\nabla \log \rho(x) - \nabla \log \pi(x))\|_2^2 \rho(x) dx. \tag{4}$$

The objective functional evaluates the least-square discrepancy between $\nabla \log \rho - \nabla \log \pi$ and $\nabla \Phi$ weighted by the density $\rho$. The optimal solution follows $\Phi = \log \rho - \log \pi$, up to a constant shift. Let $\mathcal{H} \subseteq C^1(\mathbb{R}^d)$ be a finite dimensional function space. The following proposition gives a formulation of (4) in $\mathcal{H}$.

**Proposition 1** *Let $\mathcal{H} \subseteq C^1(\mathbb{R}^d)$ be a function space. The variational problem* (4) *in the domain $\mathcal{H}$ can be reformulated to*

$$\inf_{\Phi \in \mathcal{H}} \frac{1}{2} \int \|\nabla \Phi(x)\|_2^2 \rho dx + \int \Delta \Phi(x) \rho(x) dx + \int \langle \nabla \log \pi(x), \nabla \Phi(x) \rangle \, \rho(x) dx. \tag{5}$$

**Remark 1** A similar variational problem has been studied in (di Langosco et al., 2021). If we replace $\nabla \Phi$ for $\Phi \in \mathcal{H}$ by a vector field $\Psi$ in certain function family, then, the quantity in (5) is the negative regularized Stein discrepancy defined in (di Langosco et al., 2021) between $\rho$ and $\pi$ based on $\Psi$. This problem is also similar to the varitional problem for the score matching estimator in (Hyvärinen & Dayan, 2005) by parameterizing $\Phi$ in a given probabilistic model. In comparison, our method can be viewed as a special case of score matching by using a two-layer neural network.

Therefore, by replacing the density $\rho$ by finite samples $\{x_n\}_{n=1}^N \sim \rho$, the problem (5) in terms of finite samples forms

$$\inf_{\Phi \in \mathcal{H}} \frac{1}{N} \sum_{n=1}^N \left( \frac{1}{2} \|\nabla \Phi(x_n)\|_2^2 + \Delta \Phi(x_n) \right) + \frac{1}{N} \sum_{n=1}^N \langle \nabla \log \pi(x_n), \nabla \Phi(x_n) \rangle. \tag{6}$$

## 3 Optimal neural network approximation of Wasserstein gradient

In this section, we focus on functional space $\mathcal{H}$ of functions represented by two-layer neural networks. We derive the primal and dual problem of the regularized Wasserstein variational problems.

By analyzing the dual constraints, a convex SDP relaxation of the dual problem is obtained. We also present a practical implementation estimation of $\nabla \log \rho - \nabla \log \pi$ and discuss the choice of the regularization parameter.

Let $\psi$ be an activation function. Consider the case where $\mathcal{H}$ is a class of two-layer neural network with the activation function $\psi(x)$:

$$\mathcal{H} = \left\{ \Phi_{\boldsymbol{\theta}} \in C^1(\mathbb{R}^d) | \Phi_{\boldsymbol{\theta}}(x) = \alpha^T \psi(W^T x) \right\}, \tag{7}$$

where $\boldsymbol{\theta} = (W, \alpha)$ is the parameter in the neural network with $W \in \mathbb{R}^{d \times m}$ and $\alpha \in \mathbb{R}^m$.

**Remark 2** We can extend this model to handle the bias term by add an entry of 1 in $x_1, \ldots, x_n$.

For two-layer neural networks, we can compute the gradient and Laplacian of $\Phi \in \mathcal{H}$ as follows:

$$\nabla \Phi_{\boldsymbol{\theta}}(x) = \sum_{i=1}^m \alpha_i w_i \psi'(w_i^T x) = W(\psi'(W^T x) \circ \alpha), \tag{8}$$

$$\Delta \Phi_{\boldsymbol{\theta}}(x) = \sum_{i=1}^m \alpha_i \|w_i\|_2^2 \psi''(w_i^T x). \tag{9}$$

Here $\circ$ represents the element-wise multiplication. By adding a regularization term to the variational problem (6), we obtain

$$\min_{\boldsymbol{\theta}} \frac{1}{2N} \sum_{n=1}^N \left\| \sum_{i=1}^m \alpha_i w_i \psi'(w_i^T x_n) \right\|_2^2 + \frac{1}{N} \sum_{n=1}^N \left\langle \sum_{i=1}^m \alpha_i w_i \psi'(w_i^T x_n), \nabla \log \pi(x_n) \right\rangle$$
$$+ \frac{1}{N} \sum_{n=1}^N \sum_{i=1}^m \alpha_i \|w_i\|_2^2 \psi''(w_i^T x_n) + \frac{\beta}{2} R(\boldsymbol{\theta}), \tag{10}$$

where $\beta > 0$ is the regularization parameter. We focus on the squared ReLU activation $\psi(z) = (z)_+^2 = (\max\{z, 0\})^2$. Note that a non-vanishing second derivative is required for the Laplacian term in (9), which makes the ReLU activation inadequate. For this activation function, we consider the regularization function $R(\boldsymbol{\theta}) = \sum_{i=1}^m (\|w_i\|_2^3 + |\alpha_i|^3)$.

**Remark 3** We note that $\nabla \Phi_{\boldsymbol{\theta}}(x)$ and $\Delta \Phi_{\boldsymbol{\theta}}(x)$ are all piece-wise degree-3 polynomials of the parameters $\boldsymbol{\theta}$. Hence, we consider a specific cubic regularization term above, analogous to (Bartan & Pilanci, 2021a). By choosing this regularization term, we can derive a simplified dual problem.

By utilizing the arithmetic and geometric mean (AM-GM) inequality, we can rescale the first and second-layer parameters and formulate the regularized variational problem (10) as follows.

**Proposition 2 (Primal problem)** *The regularized variational problem* (10) *can be reformulated to*

$$\min_{W, \alpha} \frac{1}{2} \sum_{n=1}^N \left\| \sum_{i=1}^m \alpha_i w_i \psi'(w_i^T x_n) \right\|^2 + \sum_{n=1}^N \sum_{i=1}^m \alpha_i \|w_i\|_2^2 \psi''(w_i^T x_n)$$
$$+ \sum_{n=1}^N \left\langle \sum_{i=1}^m \alpha_i w_i \psi'(w_i^T x_n), \nabla \log \pi(x_n) \right\rangle + \tilde{\beta} \|\alpha\|_1, \tag{11}$$
$$s.t. \ \|w_i\|_2 \le 1, i \in [m],$$

*where* $\tilde{\beta} = 3 \cdot 2^{-5/3} N \beta$ *and we denote* $[m] = \{1, \ldots, m\}$.

In short, the optimal value of (10) and (11) are the same. We can obtain the optimal solution of (11) by rescaling the optimal solution of (10) and vice versa. For simplicity, we write $Y \in \mathbb{R}^{N \times d}$ whose $n$-row is $\nabla \log \pi(x_n)$ for $n \in [N]$. We introduce the slack variable $z_n = \sum_{i=1}^m \alpha_i w_i \psi'(x_n^T w_i)$ for $n \in [N]$ and denote $Z = [z_1 \ \cdots \ z_N]^T \in \mathbb{R}^{N \times d}$. Then, we can simplify the problem (11) to

$$\min_{W, \alpha, Z} \frac{1}{2} \|Z\|_F^2 + \sum_{n=1}^N \sum_{i=1}^m \alpha_i \|w_i\|_2^2 \psi''(w_i^T x_n) + \operatorname{tr}(Y^T Z) + \tilde{\beta} \|\alpha\|_1, \tag{12}$$
$$s.t. \ z_n = \sum_{i=1}^m \alpha_i w_i \psi'(x_n^T w_i), n \in [N], \|w_i\|_2 \le 1, i \in [m].$$

To derive the convex relaxtion of neural network training problem, the dual problem plays an import role. By applying the Lagrangian duality, we can derive the dual problem of (12) as follows.

**Proposition 3 (Dual problem)** *The dual problem of the regularized variational problem* (12) *is*

$$\max_{\Lambda} \ -\frac{1}{2}\|\Lambda + Y\|_F^2, \ s.t. \ \max_{w:\|w\|_2 \leq 1} \left| \sum_{n=1}^{N} \|w\|_2^2 \psi''(x_n^T w) - \lambda_n^T w \psi'(x_n^T w) \right| \leq \tilde{\beta}, \tag{13}$$

*which provides a lower-bound on* (12).

We note that the dual problem can be infeasible if the regularization parameter $\tilde{\beta}$ is below certain threshold. In other words, if the regularization term is missing or the regularization parameter is not large enough, the optimal value of the dual problem is $-\infty$ and the primal problem is not lower bounded.

### 3.1 ANALYSIS OF DUAL CONSTRAINTS AND THE RELAXED DUAL PROBLEM

Now, we analyze the constraint in the dual problem. We note that it is closely related to the regularization parameter, which we will discuss later. For simplicity, we take $\psi''(0) = 0$ as the subgradient of $\psi'(z)$ at $z = 0$, i.e., taking the left derivative of $\psi'(z)$ at $z = 0$. Let $X = [x_1, \ldots, x_N]^T \in \mathbb{R}^{N \times d}$. Denote the set of all possible hyper-plane arrangements corresponding to the rows of $X$ as

$$\mathcal{S} = \{\mathbf{diag}(\mathbb{I}(Xw \geq 0))|w \in \mathbb{R}^d, w \neq 0\}. \tag{14}$$

Here $\mathbb{I}(s) = 1$ if the statement $s$ is correct and $\mathbb{I}(s) = 0$ otherwise. Let $p = |\mathcal{S}|$ be the cardinality of $\mathcal{S}$, and write $\mathcal{S} = \{D_1, \ldots, D_p\}$. According to (Cover, 1965), we have the upper bound $p \leq 2r \left(\frac{e(N-1)}{r}\right)^r$, where $r = \text{rank}(X)$. Based on the analysis of the dual constraints, we can derive a convex SDP as a relaxed dual problem.

**Proposition 4 (Relaxed dual problem)** *The relaxed dual problem is the following SDP:*

$$\max_{\Lambda, \{r^{(j,-)}, r^{(j,+)}\}_{j=1}^p} -\frac{1}{2}\|\Lambda + Y\|_F^2,$$

$$s.t. \ \tilde{A}_j(\Lambda) + \tilde{B}_j + \sum_{n=0}^{N} r_n^{(j,-)} H_n^{(j)} + \tilde{\beta} e_{d+1} e_{d+1}^T \succeq 0, r^{(j,-)} \geq 0, j \in [p], \tag{15}$$

$$-\tilde{A}_j(\Lambda) - \tilde{B}_j + \sum_{n=0}^{N} r_n^{(j,+)} H_n^{(j)} + \tilde{\beta} e_{d+1} e_{d+1}^T \succeq 0, r^{(j,+)} \geq 0, j \in [p],$$

*where we denote* $[p] = \{1, \ldots, p\}$. *For* $j \in [p]$, *we denote* $A_j(\Lambda) = -\Lambda^T D_j X - X^T D_j \Lambda$, $B_j = 2 \text{tr}(D_j) I_d$, $\tilde{A}_j(\Lambda) = \begin{bmatrix} A_j(\Lambda) & 0 \\ 0 & 0 \end{bmatrix}$, $\tilde{B}_j = \begin{bmatrix} B_j & 0 \\ 0 & 0 \end{bmatrix}$, $H_0^{(j)} = \begin{bmatrix} I_d & 0 \\ 0 & -1 \end{bmatrix}$ *and* $H_n^{(j)} = \begin{bmatrix} 0 & (1 - 2(D_j)_{nn})x_n \\ (1 - 2(D_j)_{nn})x_n^T & 0 \end{bmatrix}$, $n \in [N]$ *The vector* $e_{d+1} \in \mathbb{R}^{d+1}$ *satisfies that* $(e_{d+1})_i = 0$ *for* $i \in [d]$ *and* $(e_{d+1})_{d+1} = 1$.

*The optimal value of* (15) *gives a lower bound on the dual problem* (13), *and hence on the primal problem* (12).

The relaxed bi-dual problem provides insights on approximating the primal problem via convex optimization, which is derived as follows. As an equivalent formulation of the convex dual problem (15), it can be viewed as a convex relaxation of the primal problem (12).

**Proposition 5 (Relaxed bi-dual problem)** *The dual of the relaxed dual problem* (15) *is as follows*

$$
\min_{Z,\{(S^{(j,+)},S^{(j,-)})\}_{j=1}^p} \frac{1}{2}\|Z+Y\|_F^2 - \frac{1}{2}\|Y\|_F^2 + \sum_{j=1}^p \operatorname{tr}(\tilde{B}_j(S^{(j,+)} - S^{(j,-)}))
$$

$$
+ \tilde{\beta} \sum_{j=1}^p \operatorname{tr}\left((S^{(j,+)} + S^{(j,-)})e_{d+1}e_{d+1}^T\right), \tag{16}
$$

$$
s.t. \ Z = \sum_{j=1}^p \tilde{A}_j^*(S^{(j,-)} - S^{(j,+)}),
$$

$$
\operatorname{tr}(S^{(j,-)}H_n^{(j)}) \leq 0, \operatorname{tr}(S^{(j,+)}H_n^{(j)}) \leq 0, n = 0, \ldots, N, j \in [p].
$$

*Here $A_j^*$ is the adjoint operator of the linear operator $A_j$.*

As (15) is a convex problem and the Slater's condition is satisfied, the optimal values of (15) and (16) are same. The bi-dual problem (16) is closely related to the primal problem (12). Indeed, any feasible solutions of the primal problem (11) can be mapped to feasible solutions of (16). We note that the mapping from the primal solution to the bi-dual solution cannot go both ways, unless these two problems are equivalent.

**Theorem 1** *Suppose that $(Z, W, \alpha)$ is feasible to the primal problem* (12). *Then, there exist matrices $\{S^{(j,+)}, S^{(j,-)}\}_{j=1}^p$ constructed from $(W, \alpha)$ such that $(Z, \{S^{(j,+)}, S^{(j,-)}\}_{j=1}^p)$ is feasible to the relaxed bi-dual problem* (16). *Moreover, the objective value of the relaxed bi-dual problem* (16) *at $(Z, \{S^{(j,+)}, S^{(j,-)}\}_{j=1}^p)$ is the same as objective value of the primal problem* (12) *at $(Z, W, \alpha)$.*

Let $J(Z, \{S^{(j,+)}, S^{(j,-)}\}_{j=1}^p)$ denote the objective value of the relaxed bi-dual problem (16) at a feasible solution $(Z, \{S^{(j,+)}, S^{(j,-)}\}_{j=1}^p)$. Let $(Z^*, W^*, \alpha^*)$ denote a globally optimal solution of the primal problem (12). By Theorem 1, there exist matrices $\{S^{(j,+)}, S^{(j,-)}\}_{j=1}^p$ such that $(Z^*, \{S^{(j,+)}, S^{(j,-)}\}_{j=1}^p)$ is a feasible solution of the relaxed bi-dual problem (16) and $J(Z^*, \{S^{(j,+)}, S^{(j,-)}\}_{j=1}^p)$ is the same as the objective value of (12) at its global minimum $(Z^*, W^*, \alpha^*)$. On the other hand, let $(\tilde{Z}^*, \{\tilde{S}^{(j,+)}, \tilde{S}^{(j,-)}\}_{j=1}^p)$ denote an optimal solution of the relaxed bi-dual problem (16). From the optimality of $(\tilde{Z}^*, \{\tilde{S}^{(j,+)}, \tilde{S}^{(j,-)}\}_{j=1}^p)$, we have

$$
J(\tilde{Z}^*, \{\tilde{S}^{(j,+)}, \tilde{S}^{(j,-)}\}_{j=1}^p) \leq J(Z^*, \{S^{(j,+)}, S^{(j,-)}\}_{j=1}^p). \tag{17}
$$

Note that at $(Z^*, W^*, \alpha^*)$ we obtain the optimal approximation of $\nabla \log \rho - \nabla \log \pi$ at $x_1, \ldots, x_N$ in the family of two-layer squared-ReLU networks (7). Smaller or equal objective value of the relaxed bi-dual problem (16) can be achieved at $(\tilde{Z}^*, \{\tilde{S}^{(j,+)}, \tilde{S}^{(j,-)}\}_{j=1}^p)$ than at $(Z^*, \{S^{(j,+)}, S^{(j,-)}\}_{j=1}^p)$. Therefore, we can view $\tilde{Z}^*$ gives an optimal approximation of $\nabla \log \rho - \nabla \log \pi$ evaluated on $x_1, \ldots, x_N$ in a broader function family including the two-layer squared ReLU neural networks.

From the derivation of the relaxed bi-dual problem, we have the relation $\tilde{Z}^* = -\Lambda^* - Y$, where $(\Lambda^*, \{r^{(j,+)}, r^{(j,-)})$ is optimal to the relaxed dual problem (15) and $(\tilde{Z}^*, \{\tilde{S}^{(j,+)}, \tilde{S}^{(j,-)}\}_{j=1}^p)$ is optimal to the relaxed bi-dual problem (16). Therefore, by solving $\Lambda^*$ from the relaxed dual problem (15), we can use $-\Lambda^* - Y$ as the approximation of $\nabla \log \rho - \nabla \log \pi$ evaluated on $x_1, \ldots, x_N$.

**Remark 4** We note that solving the proposed convex optimization problem 15 renders the approximation of the Wasserstein gradient direction. Compared to the two-layer ReLU networks, it induces a broader class of functions represented by $\{S^{(j,+)}, S^{(j,-)}\}_{j=1}^p$. This contains more variables than the neural network function.

## 3.2 PRACTICAL IMPLEMENTATION

Although the number $p$ of all possible hyper-plane arrangements is upper bounded by $2r((N-1)e/r)^r$ with $r = \operatorname{rank}(X)$, it is computationally costly to enumerate all possible $p$ matrices $D_1, \ldots, D_p$ to represent the constraints in the relaxed dual problem (4). In practice, we first randomly sample $M$ i.i.d. random vectors $u_1, \ldots, u_M \sim \mathcal{N}(0, I_d)$ and generate a subset

$\hat{\mathcal{S}} = \{\mathbf{diag}(\mathbb{I}(Xu_j \geq 0)|j \in [M]\}$. of $\mathcal{S}$. Then, we optimize the randomly sub-sampled version of the relaxed dual problem based on the subset $\hat{\mathcal{S}}$ and obtain the solution $\Lambda$. Here $-\Lambda - Y$ is used as the direction to update the particle system $X$. If the regularization parameter is too large, then we will have $-\Lambda - Y = 0$, which makes the particle system unchanged. Therefore, to ensure that $\tilde{\beta}$ is not too large, we decay $\tilde{\beta}$ by a factor $\gamma_1 \in (0, 1)$. This also appears in (Ergen et al., 2021). On the other hand, if $\tilde{\beta}$ is too small resulting the relaxed dual problem (4) infeasible, we increase $\tilde{\beta}$ by multiplying $\gamma_2^{-1}$, where $\gamma_2 \in (0, 1)$. Detailed explanation of the adjustment of the regularization parameter can be found in Appendix D. The overall algorithm is summarized in Algorithm 1.

---

**Algorithm 1** Convex neural Wasserstein descent

**Require:** initial positions $\{x_0^n\}_{n=1}^N$, step size $\alpha_l$, initial regularization parameter $\tilde{\beta}_0, \gamma_1, \gamma_2 \in (0, 1)$.

1: **while** not converge **do**
2:     Form $X_l$ and $Y_l$ based on $\{x_l^n\}_{n=1}^N$ and $\{\nabla \log \pi(x_l^n)\}_{n=1}^N$.
3:     Solve $\Lambda_l$ from the relaxed dual problem (15) with $\tilde{\beta} = \tilde{\beta}_l$.
4:     **if** the relaxed dual problem with $\tilde{\beta} = \tilde{\beta}_l$ is infeasible **then**
5:         Set $X_{l+1} = X_l$ for $n \in [N]$ and set $\tilde{\beta}_{l+1} = \gamma_2^{-1}\tilde{\beta}_l$.
6:     **else**
7:         Update $X_{l+1} = X_l + \alpha_l(\Lambda_l + Y_l)$ for $n \in [N]$ and set $\tilde{\beta}_{l+1} = \gamma_1\tilde{\beta}_l$.
8:     **end if**
9: **end while**

---

Applying the standard interior point method (Boyd et al., 2004) leads to the computational time

$$O((\max\{N, d^2\}\hat{p})^6). \tag{18}$$

For high-dimensional problems, i.e., $d$ is large, the computational cost of solving (15) can be large. In this case, we apply the dimension-reduction techniques (Zahm et al., 2018; Chen & Ghattas, 2020; Wang et al., 2021a) to reduce the parameter dimension $d$ to a data-informed intrinsic dimension $\hat{d}$, which is often very low, i.e., $\hat{d} \ll d$, which can dramatically decrease the computational time (18).

## 4 NUMERICAL EXPERIMENTS

In this section, we present numerical results to compare WGD approximated by neural networks (WGD-NN) and WGD approximated using convex optimization formulation of neural networks (WGD-cvxNN). The performance of compared methods is assessed by the sample goodness-of-fit of the posterior. For WGD-NN, in each iteration, it updates the particle system using (3) with a function $\Phi$ represented by a two-layer squared ReLU neural network. The parameters of the neural network is obtained by directly solving the nonconvex optimization problem (10). For high-dimensional problems, we apply the dimension reduction technique and compare the projected versions (pWGD-NN and pWGD-cvxNN).

We note that although the cost for solving the relaxed dual problem (15) using standard convex optimization solvers in WGD-cvxNN can be higher compared to that by a direct neural network training in WGD-NN, this cost difference is negligible in the entire optimization dominated by the likelihood evaluation when the model (e.g., PDE) is expensive to solve. In such cases WGD-cvxNN and WGD-NN have similar computational complexity but WGD-cvxNN achieves better performance. We use the standard convex optimization solver CVXPY (Diamond & Boyd, 2016) with MOSEK(ApS, 2019) inner solver. Applying randomized SDP solvers (Yurtsever et al., 2021), randomized second-order methods (Pilanci & Wainwright, 2017; Lacotte et al., 2021) or advanced SDP solvers (Zhao et al., 2010; Yang et al., 2015; Wang et al., 2021b) for large-scale problem can improve the computation time. Moreover, the induced SDPs have specific structures of many similar constraints. Solving the SDP (15) can be accelerated by designing a specialized convex optimization solver, which is left for future work.

### 4.1 A TOY EXAMPLE

We test the performance of WGD on a bimodal 2-dimensional double-banana posterior distribution introduced in (Detommaso et al., 2018). We first generate 300 posterior samples by a Stein variational Newton (SVN) method (Detommaso et al., 2018) as the reference, as shown in Figure 1. We evaluate the performance of WGD-NN and WGD-cvxNN by calculating the maximum mean discrepancy (MMD) between their samples in each iteration and the reference samples. In the comparison, we use $N = 50$ samples and run for 100 iterations with step sizes $\alpha_l = 10^{-3}$. For WGD-cvxNN, we set $\beta = 1$, $\gamma_1 = 0.95$ and $\gamma_2 = 0.95^{10}$. For WGD-NN, we use $m = 200$ neurons and optimize the regularized training problem (10) using all samples with the Adam optimizer (Kingma & Ba, 2014) with learning rate $10^{-3}$ for 200 sub-iterations. We also set the regularization parameter $\beta = 1$ and decrease it by a factor of 0.95 in each iteration. We find that this setup of parameters is more suitable.

The posterior density and the sample distributions by WGD-cvxNN and WGD-NN at the final step of 100 iterations are shown in Figure 1. It can be observed that WGD-cvxNN provides more representative samples than WGD-NN for the posterior density. In Figure 2, we plot the MMD of the samples by WGD-cvxNN and WGD-NN compared to the reference SVN samples at each iteration. We observe that the samples by WGD-cvxNN achieves much smaller MMD than those of WGD-NN compared to the reference SVN samples, which is consistent with the results shown in Figure 1.

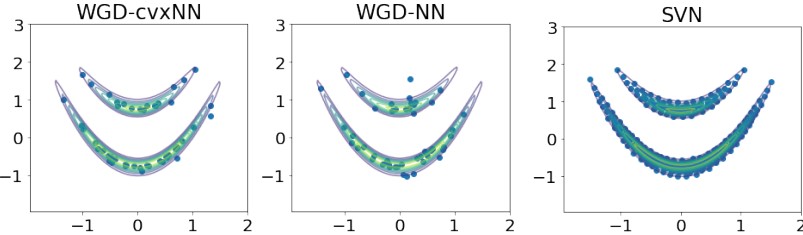

Figure 1: Posterior density and sample distributions by WGD-cvxNN and WGD-NN at the final step of 100 iterations, compared to the reference SVN samples (right).

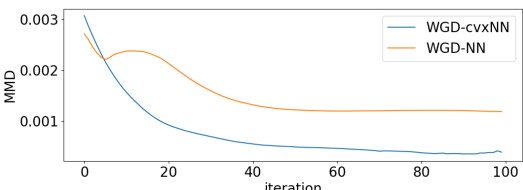

Figure 2: MMD of WGD-cvxNN and WGD-NN samples compared to the reference SVN samples.

### 4.2 PDE-CONSTRAINED NONLINEAR BAYESIAN INFERENCE

In this experiment, we consider a nonlinear Bayesian inference problem constrained by the following partial differential equation (PDE) (Chen & Ghattas, 2020) with application to subsurface (Darcy) flow in a physical domain $D = (0, 1)^2$,

$$
\begin{aligned}
\mathbf{v} + e^x \nabla u &= 0 \quad \text{in } D, \\
\nabla \cdot \mathbf{v} &= h \quad \text{in } D,
\end{aligned}
\tag{19}
$$

where $u$ is pressure, $\mathbf{v}$ is velocity, $h$ is force, $e^x$ is a random (permeability) field equipped with a Gaussian prior $x \sim \mathcal{N}(x_0, C)$ with covariance operator $C = (-\delta\Delta + \gamma I)^{-\alpha}$ where we set $\delta = 0.1, \gamma = 1, \alpha = 2$ and $x_0 = 0$. This problem is widely used in many areas, for instance, estimating permeability in groundwater flow, thermal conductivity in material science or electrical impedance in medical imaging, We impose Dirichlet boundary conditions $u = 1$ on the top boundary and $u = 0$ on the bottom boundary, and homogeneous Neumann boundary conditions on the left

and right boundaries for $u$. We use a finite element method with piecewise linear elements for the discretization of the problem, resulting in 81 dimensions for the discrete parameter. The data is generated as pointwise observation of the pressure field at 49 points equidistantly distributed in $(0, 1)^2$, corrupted with additive $5\%$ Gaussian noise. We use a DILI-MCMC algorithm Cui et al. (2016) with 10000 effective samples to compute the sample mean and sample variance, which are used as the reference values to assess the goodness of the samples.

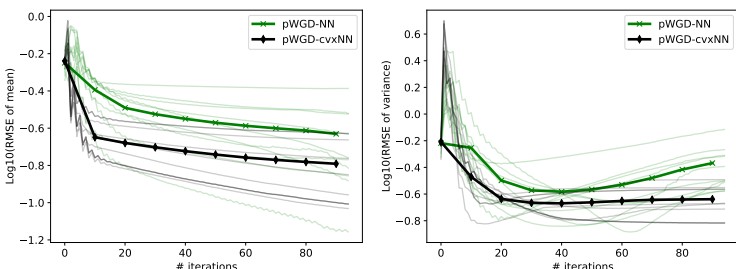

Figure 3: Ten trials and the RMSE of the sample mean (top) and sample variance (bottom) by pWGD-NN and pWGD-cvxNN at different iterations. Nonlinear inference with PDE constraint.

We run pWGD-cvxNN and pWGD-NN with 64 samples for ten trials with step size $\alpha_l = 10^{-3}$, where we set $\beta = 10$, $\gamma_1 = 0.95$, and $\gamma_2 = 0.95^{10}$ for both methods. The RMSE of the sample mean and sample variance are shown in Figure 3 for the two methods at each of the iterations. We can observe that pWGD-cvxNN achieves smaller errors for both the sample mean and the sample variance compared to pWGD-NN at each iteration. Moreover, pWGD-cvxNN provides much smaller variation of the sample mean and sample variance for the ten trials compared to pWGD-NN. Furthermore, by an effective reduction of the parameter dimension from 81 to data-informed 20 in our pWGD-cvxNN, as used and analyzed in (Zahm et al., 2018; Chen & Ghattas, 2020; Wang et al., 2021a), the time for solving the SDP is significantly reduced from about 800 seconds in average to less than 1 second (about 0.7 in average), making our pWGD-cvxNN computationally efficient.

### 4.3 BAYESIAN INFERENCE FOR COVID-19

In this experiment, we use Bayesian inference to learn the dynamics of the transmission and severity of COVID-19 from the recorded data for New York state. We use the model, parameter, and data as in Chen & Ghattas (2020). More specifically, we use a compartmental model for the modeling of the transmission and outcome of COVID-19. We take the number of hospitalized cases as the observation data to infer a social distancing parameter, a time-dependent stochastic process that is equipped with a Tanh–Gaussian prior to model the transmission reduction effect of social distancing, which becomes 96 dimensions after discretization.

We use the projected Stein variational gradient descent (pSVGD) method Chen & Ghattas (2020) as the reference to evaluate the goodness of samples. We run pWGD-cvxNN and pWGD-NN using 64 samples for 100 iterations with step size $\alpha_l = 10^{-3}$, where we set $\beta = 10$, $\gamma_1 = 0.95$, and $\gamma_2 = 0.95^{10}$ for both methods as in the last example. From Figure 4 we can observe that pWGD-cvxNN produces more consistent results than pWGD-NN compared to the reference pSVGD results, for both the sample mean and 90% credible interval, both in the inference of the social distancing parameter and in the prediction of the hospitalized cases.

## 5 CONCLUSION

In the context of Bayesian inference, we approximate Wasserstein gradient direction by the gradient of functions in the family of two-layer neural networks. We propose a convex SDP relaxation of the dual of the variational primal problem, which can be solved efficiently using convex optimization methods instead of directly training the neural network as a nonconvex optimization problem. In particular, we established that the gradient obtained by the new formulation and convex optimization is at least as good as the one approximated by functions in the family of two-layer neural networks, which is demonstrated by various numerical experiments. By stacking the two-layer neu-

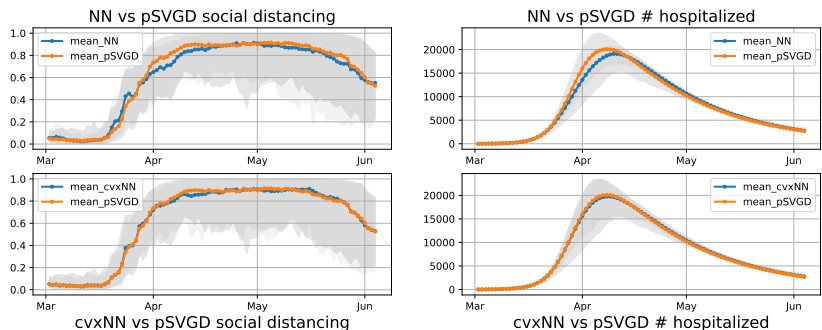

Figure 4: Comparison of pWGD-cvxNN and pWGD-NN to the reference by pSVGD for Bayesian inference of the social distancing parameter (left) from the data of the hospitalized cases (right) with sample mean and 90% credible interval.

ral networks in each step together, our proposed method formulate a deep neural network to learn the transportation map from prior to posterior. In future studies, specialized solvers for structured SDPs, including the relaxed dual problem, can lead to drastic accelerations of our proposed method and it is of central importance for the practical applications of our algorithms to real-world problems. We also expect to extend our convex optimization formulation of neural networks to the calculation/approximation of generalized Wasserstein flows. We also expect to apply deep neural networks for the approximation of Wasserstein gradient flows based on recent works on convex optimization formulations of deep neural networks (Wang et al., 2021c; Ergen & Pilanci, 2021a;b).

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

## A  CODES FOR NUMERICAL EXPERIMENT

All codes for the numerical experiment can be found in `https://github.com/ai-submit/OptimalWasserstein`.

## B  COMPARISON WITH PREVIOUS WORKS ON CONVEX OPTIMIZATION FORMULATION OF NEURAL NETWORKS

Previous works on convex optimization formulation of neural networks mainly focus on the supervised learning problem of two-layer neural networks using convex loss functions (e.g., squared loss, logistic loss). Our work utilizes a similar convex analytic framework to solve the variational problem of approximating the Wasserstein gradient direction, which is different from supervised learning. The convex optimization approach is related to the idea of infinite width neural networks modeled as probability measures. The dual problem itself is equivalent to the convex dual problem when the neural network in the primal problem has infinitely many neurons. However, the convex optimization approach tackles networks of arbitrary width that are able to learn useful representations, while the infinite width limit is quite limited (limited to basically kernel methods).

## C  ADDITIONAL NUMERICAL EXPERIMENT

### C.1  PDE-CONSTRAINED LINEAR BAYESIAN INFERENCE

In this experiment, we consider a linear Bayesian inference problem constrained by a partial differential equation (PDE) model for contaminant diffusion in environmental engineering in domain $D = (0, 1)$,

$$-\kappa \Delta u + \nu u = x \quad \text{in } D,$$

where $x$ is a contaminant source field parameter in domain $D$, $u$ is the contaminant concentration which we can observe at some locations, $\kappa$ and $\nu$ are diffusion and reaction coefficients. For simplicity, we set $\kappa, \nu = 1$, $u(0) = u(1) = 0$, and consider 15 pointwise observations of $u$ with 1% noise, equidistantly distributed in $D$. We consider a Gaussian prior distribution $x \sim \mathcal{N}(0, C)$ with covariance given by a differential operator $C = (-\delta \Delta + \gamma I)^{-\alpha}$ with $\delta, \gamma, \alpha > 0$ representing the correlation length and variance, which is commonly used in geoscience. We set $\delta = 0.1, \gamma = 1, \alpha = 1$. In this linear setting, the posterior is Gaussian with the mean and covariance given analytically, which are used as reference to assess the sample goodness. We solve this forward model by a finite element method with piece-wise linear elements on a uniform mesh of size $2^k$, $k \geq 1$. We project this high-dimensional parameter to the data-informed low dimensions as in Wang et al. (2021a) to alleviate the curse of dimensionality when applying WGD-cvxNN and WGD-NN, which we call pWGD-cvxNN and pWGD-NN, respectively. For $k = 4$ we have 17 dimensions for the discrete parameter and 4 dimensions after projection.

We run pWGD-cvxNN and pWGD-NN using 16 samples for 200 iterations with $\alpha_l = 10^{-3}$, $\beta = 5$, $\gamma_1 = 0.95$, and $\gamma_2 = 0.95^{10}$ for both methods. We use $m = 200$ neurons for pWGD-NN and train it by the Adam optimizer for 200 sub-iterations as in the first example. From Figure 5, we observe that pWGD-cvxNN achieves better root mean squared error (RMSE) than pWGD-NN for both the sample mean and the sample variance compared to the reference.

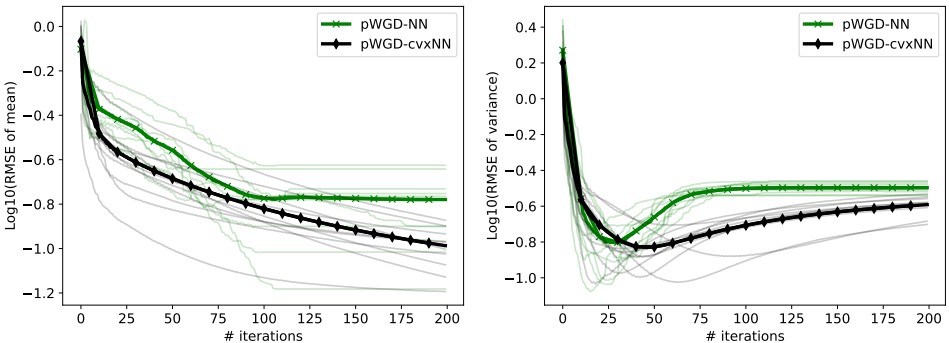

Figure 5: Ten trials and the RMSE of the sample mean (top) and sample variance (bottom) by pWGD-NN and pWGD-cvxNN at different iterations. Linear inference problem.

## D  CHOICE OF THE REGULARIZATION PARAMETER

As the constraints in the relaxed dual problem (15) depends on the regularization parameter $\tilde{\beta}$, it is possible that for small $\tilde{\beta}$, the relaxed dual problem (15) is infeasible. Consider the following SDP

$$\min \ \tilde{\beta}, \ \text{s.t.} \ \tilde{A}_j(\Lambda) + \tilde{B}_j + \sum_{n=0}^{N} r_n^{(j,-)} H_n^{(j)} + \tilde{\beta} e_{d+1} e_{d+1}^T \succeq 0,$$

$$- \tilde{A}_j(\Lambda) - \tilde{B}_j + \sum_{n=0}^{N} r_n^{(j,+)} H_n^{(j)} + \tilde{\beta} e_{d+1} e_{d+1}^T \succeq 0, \tag{20}$$

$$r^{(j,-)} \geq 0, r^{(j,+)} \geq 0, j \in [p].$$

Here the variables are $\tilde{\beta}, \Lambda$ and $\{r^{(j,+)}, r^{(j,-)}\}_{j=1}^p$. Let $\tilde{\beta}_1$ be the optimal value of the above problem. Then, only for $\tilde{\beta} \geq \tilde{\beta}_1$, there exists $\Lambda \in \mathbb{R}^{N \times d}$ satisfying the constraints in (15). In other words, the relaxed dual problem (15) is feasible. We also note that $\tilde{\beta}_1$ only depends on the samples $X$ and it does not depend on the value of $\nabla \log \pi$ evaluated on $x_1, \ldots, x_N$. On the other hand, consider the following SDP

$$\min \ \tilde{\beta}, \ \text{s.t.} \ \tilde{A}_j(Y) + \tilde{B}_j + \sum_{n=0}^{N} r_n^{(j,-)} H_n^{(j)} + \tilde{\beta} e_{d+1} e_{d+1}^T \succeq 0,$$

$$- \tilde{A}_j(Y) - \tilde{B}_j + \sum_{n=0}^{N} r_n^{(j,+)} H_n^{(j)} + \tilde{\beta} e_{d+1} e_{d+1}^T \succeq 0, \tag{21}$$

$$r^{(j,-)} \geq 0, r^{(j,+)} \geq 0, j \in [p],$$

where the variables are $\tilde{\beta}$ and $\{r^{(j,+)}, r^{(j,-)}\}_{j=1}^p$. Let $\tilde{\beta}_2$ be the optimal value of the above problem. For $\tilde{\beta} \geq \tilde{\beta}_2$, as $\mathbf{Y}$ is feasible for the constraints in (15), the optimal value of the relaxed dual problem (15) is 0. In short, only when $\tilde{\beta} \in [\tilde{\beta}_1, \tilde{\beta}_2]$, the variational problem (15) is non-trivial. To ensure that solving the relaxed dual problem (15) gives a good approximation of the Wasserstein gradient direction, we shall avoid choosing $\tilde{\beta}$ either too small or too large.

# E   PROOFS

## E.1   PROOF OF PROPOSITION 1

PROOF   We first note that

$$
\frac{1}{2} \int \|\nabla\Phi - \nabla\log\rho + \nabla\log\pi\|_2^2 \rho dx
$$
$$
= \frac{1}{2} \int \|\nabla\Phi\|_2^2 \rho dx + \int \langle \nabla\log\pi - \nabla\log\rho, \nabla\Phi \rangle \rho dx \tag{22}
$$
$$
+ \frac{1}{2} \int \|\nabla\log\rho - \nabla\log\pi\|_2^2 \rho dx.
$$

We notice that the term $\frac{1}{2}\int \|\nabla\log\rho - \nabla\log\pi\|_2^2 \rho dx$ does not depend on $\Phi$. Utilizing the integration by parts, we can compute that

$$
\int \langle \nabla\log\rho, \nabla\Phi \rangle \rho dx = \int \left\langle \frac{\nabla\rho}{\rho}, \nabla\Phi \right\rangle \rho dx
$$
$$
= \int \langle \nabla\rho, \nabla\Phi \rangle dx \tag{23}
$$
$$
= -\int \Delta\Phi \rho dx.
$$

Therefore, the variational problem (4) is equivalent to

$$
\inf_{\Phi \in C^\infty(\mathbb{R}^d)} \frac{1}{2} \int \|\nabla\Phi\|_2^2 \rho dx + \int \langle \nabla\log\pi, \nabla\Phi \rangle \rho dx + \int \Delta\Phi \rho dx. \tag{24}
$$

By restricting the domain $C^\infty(\mathbb{R}^d)$ to $\mathcal{H}$, we complete the proof.

## E.2   PROOF OF PROPOSITION 2

PROOF   Suppose that $\hat{w}_i = \beta_i^{-1} w_i$ and $\hat{\alpha}_i = \beta_i^2 \alpha_i$, where $\beta_i > 0$ is a scale parameter for $i \in [m]$. Let $\boldsymbol{\theta}' = \{(\hat{w}_i, \hat{\alpha}_i)\}_{i=1}^m$. We note that

$$
\hat{\alpha}_i \hat{w}_i \psi'(\hat{w}_i^T x_n) = \beta_i \alpha_i w_i \psi'\left(\beta_i^{-1} w_i^T x_n\right) = \alpha_i w_i \psi'(w_i^T x_n), \tag{25}
$$

and

$$
\hat{\alpha}_i \|\hat{w}_i\|_2^2 \psi''(\hat{w}_i^T x_n) = \alpha_i \|w_i\|_2^2 \psi''(\hat{w}_i^T x_n) = \alpha_i \|w_i\|_2^2 \psi''(w_i^T x_n). \tag{26}
$$

This implies that $\Phi_{\boldsymbol{\theta}}(x) = \Phi_{\boldsymbol{\theta}'}(x)$ and $\nabla \cdot \Phi_{\boldsymbol{\theta}}(x) = \nabla \cdot \Phi_{\boldsymbol{\theta}'}(x)$. For the regularization term $R(\boldsymbol{\theta})$, we note that

$$
\|\hat{w}_i\|_2^3 + \|\hat{\alpha}_i\|_2^3 = \beta_i^6 |\alpha_i|^3 + \beta_i^{-3} \|w_i\|_2^3
$$
$$
= \beta_i^6 |\alpha_i|^3 + \frac{1}{2}\beta_i^{-3}\|w_i\|_2^3 + \frac{1}{2}\beta_i^{-3}\|w_i\|_2^3 \tag{27}
$$
$$
= 3 \cdot 2^{-2/3} \|w_i\|_2^2 |\alpha_i|.
$$

The optimal scaling parameter is given by $\alpha_i = 2^{-1/9} \frac{\|w_i\|_2^{1/3}}{|\alpha_i|_1^{1/3}}$. As the scaling operation does not change $\|w_i\|_2^2 |\alpha_i|$, we can simply let $\|w_i\|_2 = 1$. Thus, the regularization term $\frac{\beta}{2} R(\boldsymbol{\theta})$ becomes $\frac{\tilde{\beta}}{N} \sum_{i=1}^m \|u_i\|_1$. This completes the proof.

### E.3 PROOF OF PROPOSITION 3

PROOF Consider the Lagrangian function

$$
\begin{aligned}
L(Z, W, \alpha, \Lambda) =& \frac{1}{2}\|Z\|_F^2 + \sum_{n=1}^{N}\sum_{i=1}^{m}\alpha_i\|w_i\|_2^2\psi''(w_i^T x_n) + \operatorname{tr}(Y^T Z) + \tilde{\beta}\|\alpha\|_1 \\
&+ \sum_{n=1}^{N}\lambda_n^T\left(z_n - \sum_{i=1}^{m}\alpha_i w_i\psi'(x_n^T w_i)\right) \\
=& \tilde{\beta}\|\alpha\|_1 + \sum_{i=1}^{m}\alpha_i\sum_{n=1}^{N}\left(\|w_i\|_2^2\psi''(w_i^T x_n) - \lambda_n^T w_i\psi'(x_n^T w_i)\right) \\
&+ \frac{1}{2}\|Z\|_F^2 + \operatorname{tr}((Y+\Lambda)^T Z).
\end{aligned}
\tag{28}
$$

For fixed $W$, the constraints on $Z$ and $\alpha$ are linear and the strong duality holds. Thus, we can exchange the order of $\min_{Z,\alpha}$ and $\max_{\Lambda}$. Thus, we can compute that

$$
\begin{aligned}
&\min_{Z,W,\alpha}\max_{\Lambda} L(Z, W, \alpha, \Lambda) \\
=&\min_{W}\max_{\Lambda}\min_{\alpha,Z} L(Z, W, \alpha, \Lambda) \\
=&\min_{W}\max_{\Lambda}\min_{\alpha,Z} \tilde{\beta}\|\alpha\|_1 + \sum_{i=1}^{m}\alpha_i\sum_{n=1}^{N}\left(\|w_i\|_2^2\psi''(w_i^T x_n) - \lambda_n^T w_i\psi'(x_n^T w_i)\right) + \frac{1}{2}\|Z\|_F^2 + \operatorname{tr}((Y+\Lambda)^T Z) \\
=&\min_{W}\max_{\Lambda} -\frac{1}{2}\|\Lambda + Y\|_F^2 + \sum_{i=1}^{m}\mathbb{I}\left(\max_{w_i:\|w_i\|_2\leq 1}\left|\sum_{n=1}^{N}\|w_i\|_2^2\psi''(w_i^T x_n) - y_n^T w_i\psi'(x_n^T w_i)\right| \leq \tilde{\beta}\right).
\end{aligned}
\tag{29}
$$

By exchanging the order of $\min$ and $\max$, we can derive the dual problem:

$$
\begin{aligned}
&\max_{\Lambda}\min_{W} -\frac{1}{2}\|\Lambda + Y\|_F^2 + \sum_{i=1}^{m}\mathbb{I}\left(\max_{w_i:\|w_i\|_2\leq 1}\left|\sum_{n=1}^{N}\|w_i\|_2^2\psi''(w_i^T x_n) - y_n^T w_i\psi'(x_n^T w_i)\right| \leq \tilde{\beta}\right) \\
=&\max_{\Lambda} -\frac{1}{2}\|\Lambda + Y\|_F^2 \ \text{s.t.} \ \max_{w_i:\|w_i\|_2\leq 1}\left|\sum_{n=1}^{N}\|w_i\|_2^2\psi''(w_i^T x_n) - y_n^T w_i\psi'(x_n^T w_i)\right| \leq \tilde{\beta}, i \in [m] \\
=&\max_{\Lambda} -\frac{1}{2}\|\Lambda + Y\|_F^2 \ \text{s.t.} \ \max_{w:\|w\|_2\leq 1}\left|\sum_{n=1}^{N}\|w\|_2^2\psi''(w^T x_n) - y_n^T w\psi'(x_n^T w)\right| \leq \tilde{\beta}, i \in [m]
\end{aligned}
\tag{30}
$$

This completes the proof.

### E.4 PROOF OF PROPOSITION 4

PROOF Based on the hyper-plane arrangements $D_1, \ldots, D_p$, the dual constraint is equivalent to that for all $j \in [p]$,

$$
\left|2\operatorname{tr}(D_j)\|w\|_2^2 - 2w^T\Lambda^T D_j X w\right| \leq \tilde{\beta}
\tag{31}
$$

holds for all $w \in \mathbb{R}^d$ satisfying $\|w\|_2 \leq 1, (2D_j - I)Xw \geq 0$. This is equivalent to say that for all $j \in [p]$

$$
\begin{aligned}
-\tilde{\beta} \geq &\min 2\operatorname{tr}(D_j)\|w\|_2^2 - 2w^T\Lambda^T D_j X w, \\
&\text{s.t. } \|w\|_2 \leq 1, 2(D_j - I)Xw \geq 0, \\
\tilde{\beta} \leq &\max 2\operatorname{tr}(D_j)\|w\|_2^2 - 2w^T\Lambda^T D_j X w, \\
&\text{s.t. } \|w\|_2 \leq 1, 2(D_j - I)Xw \geq 0.
\end{aligned}
\tag{32}
$$

From a convex optimization perspective, the natural idea to interpret the constraint (32) is to transform the minimization problem into a maximization problem. We can rewrite the minimization problem in (32) as a trust region problem with inequality constraints:

$$\min_{w \in \mathbb{R}^d} w^T \left( B_j + A_j(\Lambda) \right) w,$$
$$\text{s.t. } \|w\|_2 \leq 1, (2D_j - I)Xw \geq 0. \tag{33}$$

As the problem (33) is a convex problem, by taking the dual of (33) w.r.t. $w$, we can transform (33) into a maximization problem. However, as (33) is a trust region problem with inequality constraints, the dual problem of (33) can be very complicated. According to (Jeyakumar & Li, 2014), the optimal value of the problem (33) is bounded by the optimal value of the following SDP

$$\min_{Z \in \mathbb{S}^{d+1}} \text{tr}((\tilde{A}_j(\Lambda) + \tilde{B}_j)Z),$$
$$\text{s.t. } \text{tr}(H_n^{(j)}Z) \leq 0, n = 0, \ldots, N, \tag{34}$$
$$Z_{d+1,d+1} = 1, Z \succeq 0.$$

from below.

**Lemma 1** *The dual problem of SDP* (34) *takes the form*

$$\max -\gamma, \; s.t. \; S = \tilde{A}_j(\Lambda) + \tilde{B}_j + \sum_{n=0}^{N} r_n H_n^{(j)} + \gamma e_{d+1} e_{d+1}^T, r \geq 0, S \succeq 0, \tag{35}$$

*in variables* $r = \begin{bmatrix} r_0 \\ \vdots \\ r_N \end{bmatrix} \in \mathbb{R}^{N+1}$ *and* $\gamma \in \mathbb{R}$.

PROOF  Consider the Lagrangian

$$L(Z, r, \gamma) = \text{tr}((\tilde{A}_j(y) + \tilde{B}_j)Z) + \sum_{n=0}^{N} r_n \text{tr}(H_n^{(j)}Z) + \gamma(\text{tr}(Ze_{d+1}e_{d+1}^T) - 1), \tag{36}$$

where $r \in \mathbb{R}_+^{N+1}$ and $\gamma \in \mathbb{R}$. By minimizing $L(Z, r, \gamma)$ w.r.t. $Z \in \mathbb{S}_+^{d+1}$, we derive the dual problem (35).

The constraints on $\Lambda$ in the dual problem (13) include that the optimal value of (34) is bounded from below by $-\tilde{\beta}$. According to Lemma 1, this constraint is equivalent to that there exist $r \in \mathbb{R}^{N+1}$ and $\gamma$ such that

$$-\gamma \geq -\tilde{\beta}, S = \tilde{A}_j(\Lambda) + \tilde{B}_j + \sum_{n=0}^{N} r_n H_n^{(j)} + \gamma e_{d+1} e_{d+1}^T, r \geq 0, S \succeq 0. \tag{37}$$

As $e_{d+1}e_{d+1}^T$ is positive semi-definite, the above condition on $\Lambda$ is also equivalent to that there exist $r \in \mathbb{R}^{N+1}$ such that

$$\tilde{A}_j(\Lambda) + \tilde{B}_j + \sum_{n=0}^{N} r_n H_n^{(j)} + \tilde{\beta} e_{d+1} e_{d+1}^T \succeq 0, r \geq 0. \tag{38}$$

Therefore, the following convex set of $\Lambda$

$$\left\{ \Lambda : \tilde{A}_j(\Lambda) + \tilde{B}_j + \sum_{n=0}^{N} r_n^{(j,-)} H_n^{(j)} + \tilde{\beta} e_{d+1} e_{d+1}^T \succeq 0, \; r^{(j,-)} \geq 0 \right\} \tag{39}$$

is a subset of the set of $\Lambda$ satisfying the dual constraints

$$\left\{ \Lambda : \min_{\|w\|_2 \leq 1, (2D_j-I)w \geq 0} w^T \left( B_j + A_j(\Lambda) \right) w \geq -\tilde{\beta} \right\} \tag{40}$$

On the other hand, the constraint on $\Lambda$

$$\max_{\|w\|_2 \le 1, (2D_j - I)w \ge 0} w^T (B_j + A_j(\Lambda)) w \le \tilde{\beta} \tag{41}$$

is equivalent to

$$\min_{\|w\|_2 \le 1, (2D_j - I)w \ge 0} -w^T (B_j + A_j(\Lambda)) w \ge -\tilde{\beta}. \tag{42}$$

By applying the previous analysis on the above trust region problem, the following convex set of $\Lambda$

$$\left\{ \Lambda : -\tilde{A}_j(\Lambda) - \tilde{B}_j + \sum_{n=0}^{N} r_n^{(j,+)} H_n^{(j)} + \tilde{\beta} e_{d+1} e_{d+1}^T \succeq 0, \; r^{(j,+)} \ge 0 \right\} \tag{43}$$

is a subset of the set of $\Lambda$ satisfying the dual constraints

$$\left\{ \Lambda : \max_{\|w\|_2 \le 1, (2D_j - I)w \ge 0} w^T (B_j + A_j(\Lambda)) w \le \tilde{\beta} \right\}. \tag{44}$$

Therefore, replacing the dual constraint $\max_{w:\|w\|_2 \le 1} \left| \sum_{n=1}^{N} \|w\|_2^2 \psi''(w^T x_n) - y_n^T w \psi'(x_n^T w) \right| \le \tilde{\beta}$ by

$$\tilde{A}_j(\Lambda) + \tilde{B}_j + \sum_{n=0}^{N} r_n^{(j,-)} H_n^{(j)} + \tilde{\beta} e_{d+1} e_{d+1}^T \succeq 0, j \in [p],$$
$$- \tilde{A}_j(\Lambda) - \tilde{B}_j + \sum_{n=0}^{N} r_n^{(j,+)} H_n^{(j)} + \tilde{\beta} e_{d+1} e_{d+1}^T \succeq 0, j \in [p], \tag{45}$$
$$r^{(j,-)} \ge 0, r^{(j,+)} \ge 0, j \in [p].$$

we obtain the relaxed dual problem. As its feasible domain is a subset of the feasible domain of the dual problem, the optimal value of the relaxed dual problem gives a lower bound for the optimal value of the dual problem.

### E.5 PROOF OF PROPOSITION 5

PROOF Consider the Lagrangian function

$$L(\Lambda, \mathbf{r}, \mathbf{S}) = -\frac{1}{2} \|\Lambda + Y\|_2^2 - \sum_{j=1}^{p} \text{tr} \left( S^{(j,-)} \left( \tilde{A}_j(\Lambda) + \tilde{B}_j + \sum_{n=0}^{N} r_n^{(j,-)} H_n^{(j)} + \frac{\tilde{\beta}}{2} e_{d+1} e_{d+1}^T \right) \right)$$
$$- \sum_{j=1}^{p} \text{tr} \left( S^{(j,+)} \left( -\tilde{A}_j(\Lambda) - \tilde{B}_j + \sum_{n=0}^{N} r_n^{(j,+)} H_n^{(j)} + \frac{\tilde{\beta}}{2} e_{d+1} e_{d+1}^T \right) \right), \tag{46}$$

where we write

$$\mathbf{r} = \left( r^{(1,-)}, \dots, r^{(p,-)}, r^{(1,+)}, \dots, r^{(p,+)} \right) \in \left( \mathbb{R}^{N+1} \right)^{2p},$$
$$\mathbf{S} = \left( S^{(1,-)}, \dots, S^{(p,-)}, S^{(1,+)}, \dots, S^{(p,+)} \right) \in \left( \mathbb{S}_+^{d+1} \right)^{2p}. \tag{47}$$

Here we write $\mathbb{S}_+^{d+1} = \{ S \in \mathbb{S}^{d+1} | S \succeq 0 \}$. By maximizing w.r.t. $\Lambda$ and $\mathbf{r}$, we derive the bi-dual problem (16).

### E.6 PROOF OF THEOREM 1

Suppose that $(Z, W, \alpha)$ is a feasible solution to (11). Let $D_{j_1}, \dots, D_{j_k}$ be the enumeration of $\{\mathbf{diag}(\mathbb{I}(Xw_i \ge 0)) | i \in [m]\}$. For $i \in [k]$, we let

$$S^{(j_i,+)} = \sum_{l:\alpha_l \ge 0, \mathbf{diag}(\mathbb{I}(Xw_l \ge 0)) = D_{j_i}} \alpha_l \begin{bmatrix} w_l w_l^T & w_l \\ w_l^T & 1 \end{bmatrix}, S^{(j_i,-)} = 0, \tag{48}$$

and

$$S^{(j_i,+)} = 0, S^{(j_i,-)} = - \sum_{l:\alpha_l<0,\mathbf{diag}(\mathbb{I}(Xw_l\geq0))=D_{j_i}} \alpha_l \begin{bmatrix} w_l w_l^T & w_l \\ w_l^T & 1 \end{bmatrix}. \tag{49}$$

For $j \notin \{j_1, \ldots, j_k\}$, we simply set $S^{(j,+)} = 0, S^{(j,-)} = 0$. As $\|w_i\|_2 \leq 1$ and $D_{j_i} = \mathbb{I}(Xw_i \geq 0)$, we can verify that $\mathrm{tr}(S^{(j,-)}H_n^{(j)}) \leq 0, \mathrm{tr}(S^{(j,+)}H_n^{(j)}) \leq 0$ are satisfied for $j = j_1, \ldots, j_m$ and $n = 0, 1, \ldots, N$. This is because for $n = 0$, as $H_0^{(j_i)} = \begin{bmatrix} I_d & 0 \\ 0 & -1 \end{bmatrix}$, it follows that

$$\mathrm{tr}(S^{(j_i,+)}H_0^{(j_i)}) = \sum_{l:\alpha_l\geq0,\mathbf{diag}(\mathbb{I}(Xw_l\geq0))=D_{j_i}} \alpha_l(\|w_l\|^2 - 1) \leq 0,$$
$$\mathrm{tr}(S^{(j_i,-)}H_0^{(j_i)}) = - \sum_{l:\alpha_l<0,\mathbf{diag}(\mathbb{I}(Xw_l\geq0))=D_{j_i}} \alpha_l(\|w_l\|^2 - 1) \leq 0. \tag{50}$$

For $n = 1, \ldots, N$, we have

$$\mathrm{tr}(S^{(j_i,+)}H_0^{(j_i)}) = \sum_{l:\alpha_l\geq0,\mathbf{diag}(\mathbb{I}(Xw_l\geq0))=D_{j_i}} 2\alpha_l(1 - 2(D_{j_i})_{nn})x_n^T w_l \leq 0,$$
$$\mathrm{tr}(S^{(j_i,-)}H_0^{(j_i)}) = - \sum_{l:\alpha_l<0,\mathbf{diag}(\mathbb{I}(Xw_l\geq0))=D_{j_i}} \alpha_l(1 - 2(D_{j_i})_{nn})x_n^T w_l \leq 0. \tag{51}$$

Based on the above transformation, we can rewrite the bidual problem in the form of the primal problem (12). For $S \in \mathbb{S}^{d+1}$, we note that

$$\mathrm{tr}(S\tilde{A}_j(\Lambda))$$
$$= - \mathrm{tr}((\Lambda^T D_j X + X^T D_j \Lambda)S_{1:d,1:d})$$
$$= - 2\mathrm{tr}(\Lambda^T D_j X S_{1:d,1:d}),$$

where $S_{1:d,1:d}$ denotes the $d \times d$ block of $S$ consisting the first $d$ rows and columns. This implies that $\tilde{A}_j^*(S) = -2D_j X S_{1:d,1:d}$. Hence, we have

$$\tilde{A}_{j_i}(S^{(j_i,+)} - S^{(j_i,-)}) = - \sum_{l:\mathbf{diag}(\mathbb{I}(Xw_l\geq0))} 2\alpha_l D_{j_i} X w_l w_l^T = - \sum_{l:\mathbf{diag}(\mathbb{I}(Xw_l\geq0))} 2\alpha_l(Xw_l)_+ w_l^T.$$

Therefore, we have

$$\sum_{j=1}^p \tilde{A}_j^*(S^{(j,-)} - S^{(j,+)}) = 2\sum_{i=1}^m \alpha_i(Xw_i)_+ w_i^T.$$

As $n$-th row of $Z$ satisfies that $z_n = 2\sum_{i=1}^m \alpha_i w_i(x_n^T w_i)_+$, this implies that

$$Z = 2\sum_{i=1}^m \alpha_i(Xw_i)_+ w_i^T = \sum_{j=1}^p \tilde{A}_j^*(S^{(j,-)} - S^{(j,+)}).$$

Hence $(Z, \{(S^{(j,-)}, (S^{(j,-)}\}_{j=1}^p)$ is feasible to the relaxed bi-dual problem (16).

We can also compute that

$$\sum_{j=1}^p \mathrm{tr}(\tilde{B}_j(S^{(j,+)} - S^{(j,-)})) = 2\sum_{i=1}^m \alpha_i \sum_{n=1}^N \mathbb{I}(x_n^T w_i \geq 0)\|w_i\|_2^2,$$

and

$$\sum_{j=1}^p \mathrm{tr}\left((S^{(j,+)} + S^{(j,-)})e_{d+1}e_{d+1}^T\right) = \sum_{i=1}^m |\alpha_i|.$$

Thus, the primal problem (12) with $(Z, W, \alpha)$ and the relaxed bi-dual problem (16) with $(Z, \{(S^{(j,-)}, (S^{(j,-)}\}_{j=1}^p)$ have the same objective value.

