# OpenReview forum: "Optimal Neural Network Approximation of Wasserstein Gradient Direction via Convex Optimization"
_ICLR.cc/2023/Conference — Submitted to ICLR 2023_

### Official Review · Reviewer_mWPZ · 2022-10-14

**Confidence:** 4
**Clarity, Quality, Novelty And Reproducibility:** See above
**Correctness:** 3
**Technical Novelty And Significance:** 3
**Empirical Novelty And Significance:** 1
**Recommendation:** 3

**Strength And Weaknesses:**

**Strenghts**
- This is probably the first work to cast this problem as a SDP problem
- The optimal is global (despite being a non convex problem initially)
- The paper is well-written

**Weaknesses**
- The exposition is lacking an I believe some part of the paper could be moved to the appendix to free some space to better put in perspective the related work on that matter
- Theoretical statements lacks precision.
- There is no clear advantage both theoretically and practically to not train directly the NN on the Wasserstein gradient.
- Finally, and probably the most important issue, is that the computational cost is prohibitive. It is not clear why at least there is no discussion on how to solve the SDP problem when d is large.

**Summary Of The Paper:**

This paper tackles the issue of computing Wasserstein gradient direction for 2-layers NNs thanks to a SDP approach with main properties that there is no need to train the underlying NN. The gradient direction is computed with the dual formulation of a least-square + a polynomial regularization term.

**Summary Of The Review:**

I believe the paper is based on an interesting idea but lacks clarity in its exposition, and fails to convince on the practical aspect.

---

### Official Review · Reviewer_efky · 2022-10-20

**Confidence:** 3
**Correctness:** 3
**Technical Novelty And Significance:** 2
**Empirical Novelty And Significance:** 3
**Recommendation:** 5

**Clarity, Quality, Novelty And Reproducibility:**

**Quality:**
Overall, the main idea of the paper as well as the execution is of sufficient quality.
The proposed relaxation is a reasonable and timely approach to the important problem of approximating Wasserstein gradients.
In particular, I appreciate the fact that the proposed method is demonstrated on more than a synthetic data.

**Clarity:**
The overall idea of relaxing the dual of a training problem is presented very clear and therefore the general structure of the manuscript is easy to follow. On a smaller scale some paragraphs and formulations are less clear. This is mainly due to a) linguistic errors, which can be easily fixed b) rather heavy notation and suboptimal formatting, which can also be addressed. I compiled a list of specific things that caught my eye in the review.

**Originality:**
Given the recent works on convex relaxations of neural network training problems in the context of supervised learning problems, the idea of the paper to apply it in the context of neural network approximations of the Wasserstein gradients seems not exceptional original. That being said, I like the idea and believe that demonstrating that SDP relaxations can also be used in this context is a valuable contribution.

**Strength And Weaknesses:**

**Strengths:**
* The Introduction is very well written and the overall idea of the paper presented clearly, which leads to a smooth overall read.
* I appreciate the idea of the convex SDP relaxation of the NN training problem encountered in the variational formulation of the Wasserstein gradient approximation with shallow networks.
* The empirical results show that the approach can be made effective even for larger problems and include problems with real world data.

**Weaknesses:**
* Where the introduction is written very nicely, the later sections are not always a very smooth read. This is mainly due to a) linguistic errors, which can easily be fixed b) rather heavy notation and suboptimal formatting, which can also be addressed.
* The discussion of related works in the introduction does a good job giving an overview over approximations of WGD. However, the relation to prior works on SDP relaxations of neural network training is not sufficiently well described. Some of these works are mentioned in the introduction, but their precise relation and in particular the difference of the manuscript to prior works here is not described. This would be important to add in order to make the contributions of the manuscript clear.
*  The experiments would be stronger if not only compared against other methods relying on Wasserstein gradient approximations, but rather on  Bayesian inference techniques.
* The title of the paper implies that using the proposed relaxation yields a better approximation of the Wasserstein gradient compared to solving the primal problem. This is however neither supported sufficiently well in an experiment nor a theoretical result as far as I can see.
* The theoretical aspects of the proposed method are not very well elaborated. In particular, the implications of Theorem 1 for the method are not laid out very transparently.
* The proposed method is only presented for the KL divergence as an objective. Although this is very well motivated by the application in Bayesian inference tasks, the contributions of the manuscript would be stronger, if if was formulated and analyzed in a more general setting.

**Summary Of The Paper:**

The manuscript proposes a convex relaxation for the approximation of sample based Wasserstein gradients with a KL divergence as an objective. The motivation for this is the application to Bayesian inference problems.
To derive the relaxation, the drift in the ODE describing the evolution of the particles can be characterized as the minimizer of an energy functional, which can be used as a training objective for a shallow neural network. A convex relaxation as a semidefinite program of the dual of this optimization problem is presented. Further, the dual of this  relaxation is given and used in order to show that the relaxation  provides a lower bound to the primal problem.
The benefits of the proposed method are demonstrated on three different problems, on a synthetic toy problem and on two Bayesian inference problems, which also include real world data.

**Summary Of The Review:**

The paper studies an important and timely problem and provides a nice extension of recent works on SDP relaxations to neural network approximation of Wasserstein gradients. Overall, I enjoyed reading the manuscript as the overall idea is clearly laid out. For more detailed comments on the different aspects, I would like to refer to my comments above.
Currently, I feel that the manuscript requires improvements in different aspects, which I summarize below:

Things that I find necessary to see addressed:

* *Relation to prior works on convex relaxations:* I think it is important to highlight the difference of the proposed relaxation to existing works on convex relaxations of neural network training problems.
* *Clarity:* Particularly Section 3 is not easy to digest in its current form. A lot of things can be improved by proof reading the section and fixing formatting (e.g. in Remark 4 and Lemma 4, Proposition 4 and the paragraph below Theorem 1). Concrete comments for Proposition 4: There is currently no statement made in the proposition, so it causes some confusion in its current form; the formatting should definitely be improved; ideally, the symbols should be introduced before the equation; $[p]$ notation is not introduced as a notation; there is a comma at the end of (15) and it continues with upper case; there is a fullstop missing after $[N]$; $A_j(\Lambda)$ is introduced but doesn’t appear in (15).
* *Optimality:* In its current form the title suggests evidence on the optimality of the proposed method for the approximation of Wasserstein gradients. Further, you state in the conclusion that „the gradient … is at least as good“. Currently I do not see results supporting this in the manuscript. Do you see this as a consequence of Theorem 1 or of your experiments? Theorem 1 implies that the regularized dual gives a lower bound on the primal and your experiments show that the resulting WGD converges faster. This does however not imply that the approximation of the Wasserstein gradient is more exact as far as I understand it. I think the claim of optimality needs to be supported better or needs to be adjusted accordingly.

Things I recommend to address:
* *Experiments:* It would be good to have experiments comparing the primal and dual problem in the following sense: What is the deviation of the two update directions of WGD-NN and WGD-cvxNN away from the true Wasserstein gradient. In particular, this could be an empirical proof that the approximation of the Wasserstein gradient is indeed better with the proposed method.
* *Consequences of Theorem 1:* The manuscript would benefit from an elaboration of the implications of Theorem 1. In particular, what does it say about the relation between the proposed relaxation and the primal problem?
* *More general setup:* The proposed method is only presented for the KL divergence as an objective. Although this is very well motivated by the application in Bayesian inference tasks, the contributions of the manuscript would be stronger, if if was formulated and analyzed in a more general setting.

Specific questions:
* Why do you call the proposed approximation optimal, i.e., does it satisfy any optimality criteria?
* What are the implications of Theorem 1?
* Proposition 1:
	* What happens with the boundary terms in the partial integration?
	* Why do you formulate Proposition 1 for a function space? I think it is just an alternative expression of the energy due to partial integration. Being nit-picky, later you consider the problem on a function class (so no linear structure), which might confuse people if they read „function space“ is something linear. I would rather simply say that the energy in (4) takes this form.

Further thoughts (not so important):
* The experiments could be stronger if not only compared against other methods relying on Wasserstein gradient approximations, but rather on  Bayesian inference techniques.
* *Variational formulation:* I am not entirely sure, whether I appreciate the wording here. The reason for this is that I am unsure whether (4) is the variational formulation of (2) in the sense that I am unsure, whether its Euler-Lagrange equations characterize the Wasserstein gradient. Think for example of a linear PDE $-\Delta u = f$ (lets say with zero boundary values), which is both the unique minimizer of $\frac12\int \lvert\nabla u\rvert^2dx-\int fu dx$ and of $\int \lvert \Delta u+f\rvert dx$, where the first one is called the variational formulation and the second one a residual minimization.
* *Formatting (pretty nit-picky):* Line breaks in section titles are not good; a single subsection in a section is not too nice, maybe a \paragraph does the job;

---

### Official Review · Reviewer_4MPo · 2022-10-21

**Confidence:** 2
**Correctness:** 4
**Technical Novelty And Significance:** 2
**Empirical Novelty And Significance:** 2
**Recommendation:** 5

**Clarity, Quality, Novelty And Reproducibility:**

The paper reads well and code is provided for reproducibility. The paper's novelty is limited as it applies well-known ideas to a somewhat specific (as in, not generalizable) problem.

**Strength And Weaknesses:**

Strength
======

- The idea of using a convex SDP relaxation of the dual of the variational primal problem is interesting to understand the behaviour of a particular family of two-layer neural nets.


Weaknesses
=========

- Using convex SDP relaxations is not a new idea per se: see, e.g., "Semidefinite Relaxation of Quadratic Optimization Problems" by Tom Luo *et al*.

Feedback
=======

- This sentence: "However, due to the nonlinear and nonconvex structure of neural networks, optimization algorithms
including **stochastic gradient descent may not find the global optima** of the training problem" is presented as if, in general, not finding the global optima is an issue in SGD. In training neural nets you precisely do not want to train till the global optima as that would likely mean **overfitting**.

**Summary Of The Paper:**

This paper uses an SDP relaxation for the problem of computing the Wasserstein gradient. They use numerical algorithms that make their proposed method suitable for practical scenarios involving Bayesian inference.

**Summary Of The Review:**

Overall the paper uses a straightforward technique to a problem that has limited practical applications (learning two-layers neural nets).

---

### Decision · Program_Chairs · 2023-01-20

**Decision:**

Reject

**Justification For Why Not Higher Score:**

The paper has an interesting idea. However, the paper needs to be improved before publication.

**Justification For Why Not Lower Score:**

N/A

**Metareview: Summary, Strengths And Weaknesses:**

In this paper, the authors propose a convex relaxation for the approximation of sample based Wasserstein gradients with a KL divergence as an objective. The reviewers agree that the idea of using a convex SDP relaxation of the dual of the variational primal problem is interesting. However, the paper can be further improved by adding the relation to prior works on SDP relaxations of neural network training, adding comparison with non-Wasserstein gradients methods, and improving the paper writing as suggested by the reviewer efky. Thus, the paper needs a major revision before publication.

I encourage the authors to revise the paper based on the reviewer's comments and submit it to future ML venues.

**Summary Of Ac-Reviewer Meeting:**

N/A